# Immunotherapy for Uterine Cervical Cancer

**DOI:** 10.3390/healthcare7030108

**Published:** 2019-09-17

**Authors:** Masahiro Kagabu, Takayuki Nagasawa, Daisuke Fukagawa, Hidetoshi Tomabechi, Saiya Sato, Tadahiro Shoji, Tsukasa Baba

**Affiliations:** Department of Obstetrics and Gynecology, Iwate Medical University School of Medicine, Morioka, Iwate 020-8505, Japan; tnagasaw@iwate-med.ac.jp (T.N.); dfukaga@yahoo.co.jp (D.F.); bechitomabehi@gmail.com (H.T.); sseiya@iwate-med.ac.jp (S.S.); tshoji@iwate-med.ac.jp (T.S.); babatsu@iwate-med.ac.jp (T.B.)

**Keywords:** cervical cancer, immunotherapy, immune checkpoint inhibitor

## Abstract

Cervical cancer is a malignant neoplastic disease that is the fourth most commonly occurring cancer in women worldwide. Since the introduction of angiogenesis inhibitors, treatments for recurrent and advanced cervical cancers have improved significantly in the past five years. However, the median overall survival in advanced cervical cancer is 16.8 months, with a 5-year overall survival rate of 68% for all stages, indicating that the effects of the treatment are still unsatisfactory. The development of a new treatment method is therefore imperative. Recently, in the clinical oncology field, remarkable progress has been made in immunotherapy. Immunotherapy is already established as standard therapy in some fields and in some types of cancers, and its clinical role in all areas, including the gynecology field, will change further based on the outcomes of currently ongoing clinical trials. This manuscript summarizes the results from previous clinical trials in cervical cancer and describes the ongoing clinical trials, as well as future directions.

## 1. Introduction

Cervical cancer is the fourth most common cancer in women, and the seventh most common of all human cancers. The global incidence of and mortality from cervical cancer in 2012 were approximately 528,000 and 266,000, respectively, according to the Agency for Research on Cancer (IARC) database [1]. Currently, the first-line treatment of early-stage to established cervical cancer, including intraepithelial cancer, is surgery. Radiotherapy and chemotherapy have been used to treat patients with advanced uterine cervical cancer [2], but with limited success. In the GOG204 study, the overall survival (OS) extended from 13 months to 17 months in recurrent and unresectable cervical cancer with the addition of bevacizumab to platinum-based chemotherapy [3]. However, because treatment regimens for second-line and beyond have yet to be established, new methods need to be developed.

Cancer immunotherapy is a general term for treatments that strengthen or trigger the patient’s own immune system to elicit anti-tumor effects. Cancer immunotherapy has a long history, starting with peptide vaccination and adoptive immunotherapy for cancer, though it did not show promising outcomes. However, with the recent introduction of immune checkpoint inhibitors that release the brakes of immune suppression, the evidence for immunotherapy in cancer treatment has finally become established. Clinical trials of immune checkpoint inhibitors have also been conducted in many areas, including cervical cancer. In particular, the programmed death-1(PD-1)/PD-1 ligand (PD-L1) inhibitor is highly effective in solid tumors, including cervical cancer, and has been approved by the Food and Drug Administration (FDA). This review describes the clinical studies of cancer immunotherapy to treat cervical cancer.

## 2. Immunotherapy Targeting HPV Related Gene in Cervical Cancer

High-risk HPV is essential for carcinogenesis and maintaining cancer characteristics for cervical cancer, and immunotherapy targeting HPV has been expected. The E6 and E7 viral proteins, critical in driving HPV oncogenesis and foreign to the human immune system, represent ideal targets for therapeutic cancer vaccination [4]. Many clinical trials of vaccine monotherapy have been conducted. However, the effectiveness of vaccine monotherapy has not been proven for advanced cervical cancer [4]. Clinical trials of vaccine monotherapy have been conducted against CIN3 rather than advanced cervical cancer [4]. In recent years, the development of adjuvants has progressed. Clinical trials of a combination therapy of vaccines for advanced cervical cancer such as ADXS11-001 and ISA101 have been conducted. ADXS11-001 is a live, irreversibly attenuated Listeria monocytogenes (Lm)–listeriolysin O (LLO) immunotherapy bioengineered to secrete an antigen-adjuvant fusion protein consisting of a truncated, nonhemolytic fragment of LLO fused to human HPV-16 E7 (tLLO-HPV-16 E7) [5]. Phase II study evaluated the safety and efficacy of ADXS11-001, administered with or without cisplatin, in patients with recurrent/refractory cervical cancer following prior chemotherapy and/or radiotherapy. This study showed a 12-month OS rate of approximately 35% (38 of 109 participants), as well as a median OS of 8.28 months in the axalimogene filolisbac monotherapy arm and 8.78 months in the combination arm [5]. Based on these results, a phase III study is currently planned for advanced cervical cancer (NCT02853604). ISA101 is among the most promising vaccines targeted to E6 and E7, and it consists of nine overlapping long E6 peptides and four overlapping 35-mer E7 peptides (synthetic long peptide HPV-16 vaccine), covering the complete sequence of the HPV-16 E6 and E7 oncoproteins. These long peptides effectively deliver antigens to dendritic cells, which then induce CD4+ and CD8+ T-cell responses by the HLA classes I and II presentation of the HPV-16 E6 and E7 processed epitope peptides [6]. Phase II study of combining Nivolumab and ISA101 for patients with incurable HPV 16-related cancer was conducted (NCT02426892). Twenty-four patients (cervical cancer: 1 patient) participated, and the overall response rate (ORR) was 33%, median progression-free survival (PFS) was 2.7 months, and median overall survival (OS) was 17.5 months. However, Grade 3 to 4 adverse events occurred in 2 patients [6]. Randomized clinical trials are needed to confirm this result.

## 3. Theoretical Basis for Checkpoint Inhibitors in Cervical Cancer

The immunopathology of cancer varies. Cancer types in which the main effectors are CD8^+^T cells specific to anti-tumor antigens (i.e., neoantigens) can be divided into “T cell inflamed,” where anti-tumor T cells are aggregated before treatment, and “non-T cell inflamed,” where such aggregation is absent. In T cell inflamed cancer, cytokines such as interferon (IFN)-γ, which are secreted by tumor antigen-specific T cells that recognize cancer cells, induce cancer cells and surrounding macrophages to express PD-L1, thereby suppressing CD8^+^T cells and cancer elimination [7]. Such pathology is thought to be present in a percentage of cervical cancers that develop through human papilloma virus (HPV) infection, indicating that drugs that block this PD-1/PD-L1 pathway would be beneficial [8].

Sustained infection by HPV is profoundly related to the carcinogenesis of cervical cancer. Researchers have conducted mechanistic analyses of the relationship between HPV status and PD-L1 expression in HPV-related solid tumors, primarily in head and neck squamous cell carcinoma (SCC) and cervical cancer. In HPV-related head and neck SCC, PD-L1 expression on the cell membrane and IFN-γ mRNA upregulation were observed. This result indicated that IFN-γ was secreted through initial HPV infection, and subsequently induced PD-L1 expression [9]. Moreover, PD-L1 DNA methylation was negatively correlated with PD-L1 mRNA expression and was significantly associated with HPV infection in The Cancer Genome Atlas (TCGA) cohorts, indicating that PD-L1 DNA methylation is associated with transcriptional silencing and HPV infection in head and neck SCC [10]. Several teams that examined whether HPV infection affects PD-L1 expression in cervical cancer found that being HPV-positive is positively correlated with increased PD-L1 expression [11,12].

PD-L1 is expressed on the surface of cervical cancer tumor cells, antigen presenting cells, and tumor-infiltrating lymphocytes (TILs), whereas most PD-1-positive cells have been identified as T cells in the stroma of cervical cancer. PD-1 expression in the tumor stroma in cervical cancer is observed in 60.82% (59/97) of patients, according to one study [13], and in 46.97% (31/66) according to another study [14]. Many researchers have also examined PD-L1 expression in cervical cancer tissue. PD-L1 expression is observed in 34.4–96% of cervical cancer tissues, but it is rarely observed in histologically normal cervical tissue [15,16]. An analysis by histology type reported that PD-L1 expression is observed in 80% of cervical SCC [12]. According to the TCGA database, PD-L1 amplification or acquisition was observed in 22% (28/129) of patients with cervical SCC [17]. Furthermore, PD-L1 can be expressed on TILs, and this plays a role in the anti-tumor response blockade. According to a study of cervical SCC samples, the PD-L1 expression rates in cancer cells and TILs were 59.1% and 47.0%, respectively [14]. These data suggest that both PD-L1 and PD-1 are widely expressed in cervical cancer tumor cells and stroma, indicating they could be a treatment target of a PD-1/PD-L1 inhibitor. To date, several lines of evidence support the possibility of using specific biomarkers to identify early stage cervical cancer and, in this way, offer a better prognosis for the patients [18,19]. Some biomarkers were analyzed in the carcinoma in situ (CIS), microinvasive, and squamous cell carcinoma (SCC) of the uterine cervix. There was a highly significant increase in PDL1 expression and decrease in Ki-67 (each *p* < 0.001) in microinvasive cancer compared to CIS, whereas p16 and E6/E7 remained stable. As the lesion progressed to SCC, p16 and E6/E7 RNA remained strongly overexpressed with a concomitant over expression of importin-β and Ki67 [19]. These data suggest that PD-L1 may be a useful biomarker to differentiate CIS from microinvasive cancer and, thus, anti-PD-L1 therapy may inhibit the progression of CIS to the invasive stage.

Cytotoxic T-lymphocyte antigen 4 (CTLA-4) is expressed on the T cell surface and, by binding to the B7 molecule expressed on the dendritic cells, it terminates T cell activity, thereby suppressing an excessive T cell immune response. CTLA-4 is constantly expressed on regulatory T cells (Tregs). As seen when analyzing the Treg frequency in tumors of cervical cancer, patients with high Treg frequency have significantly shorter OS than patients with low Treg frequency [20], indicating that an anti-CTLA-4 antibody could be a treatment target.

## 4. Clinical Research Examining Checkpoint Inhibitors for Cervical Cancer

Since 2015, clinical trials on various checkpoint inhibitors have been conducted for cervical cancer.

### 4.1. PD-1/PD-l1 Inhibitor

KEYNOTE-028 (phase Ib study) and KEYNOTE-158 (phase II study), which investigated pembrolizumab in recurrent and unresectable cervical cancers, were conducted. In KEYNOTE-028, pembrolizumab 10 mg/kg was given every 2 weeks. Twenty-four patients participated, and the overall response rate (ORR) was 17%, 6-month progression-free survival (PFS) was 13%, and 6-month OS was 66.7%. Moreover, Grade 4 adverse events were not observed [21]. Based on these results, KEYNOTE-158 was conducted. In KEYNOTE-158, pembrolizumab 200 mg/kg was given every 3 weeks. The ORR was 12.2%. Clinical response was observed only in PD-L1 positive cases. The drug effects were dependent of PD-L1 expression in this population [22]. Based on these results, the FDA approved pembrolizumab with PD-L1 IHC 22C3 PharmDx as a companion diagnostic in recurrent and unresectable advanced cervical cancer in June 2018. Moreover, CheckMate 358 (phase I–II study) using nivolumab was conducted. In this trial, nivolumab 240 mg/kg was given every 2 weeks in virus-related tumors, including cervical cancer. The ORR was 26.3%, and the disease control rate was 70.8%. For adverse events, Grade 3/4 hyponatremia and diarrhea were observed [23]. Based on the results of these studies, pembrolizumab and nivolumab appeared useful in recurrent and unresectable advanced cervical cancers, though a longer observation period is necessary in the future. (Table 1)

### 4.2. Currently Ongoing Clinical Trials

Previously conducted studies on immune checkpoint inhibitors in cervical cancer involved a monotherapy approach. (Table 2) Ongoing or planned immune checkpoint inhibitor studies in cervical cancer consist of combination therapy, aiming to achieve a greater response rate. Specifically, these studies investigate a combination with existing methods (radiation therapy or chemotherapy) or combination therapy with other molecularly targeted drugs.

Of these methods, the combination of radiation therapy and immune checkpoint inhibitors has been attracting attention. (Table 3) Concurrent chemoradiation therapy (CCRT) is the standard for treating locally advanced cervical cancer. The activation of anti-tumor immunity through radiation therapy is called the “abscopal effect.” It is the phenomenon in which the anti-tumor immunity is activated through local radiation therapy, and distant metastatic lesions subsequently shrink or disappear. Until recently, the detailed mechanism for this effect was unknown. Recent studies showed that tumor cells impaired or destroyed by radiation therapy secrete new tumor antigens, and intracellular nuclear factor κβ is activated due to DNA damage, resulting in IFN secretion. These activate the antigen-presenting cells, leading to the activation of anti-tumor immunity [24]. In unresectable lung cancer, the combination of CCRT and an immune checkpoint inhibitor was investigated, demonstrating positive effects [25]. In a phase III trial that treated patients with durvalumab after CCRT, the median PFS was 17.2 months, which was significantly longer than placebo at 5.6 months [26]. Based on these findings, the FDA approved durvalumab in unresectable lung cancer. However, no conclusions have been reached about the timing of this drug, or whether it is better as a combination or as maintenance therapy [25]. Based on this, clinical trials have been initiated in cervical cancer patients receiving CCRT.

Some ongoing trials are testing the addition of immune checkpoint inhibitors to systemic chemotherapy (paclitaxel/carboplatin or cisplatin with bevacizumab) in unresectable or recurrent advanced cervical cancer. (Table 4, Table 5 and Table 6).

## 5. Development of Novel Immunotherapy

### 5.1. Genetically Modified T Cell Transfer Therapies

In this method, antigen-specific receptors are artificially expressed on the T cell surface. These T cells are cultured in large numbers outside the body and subsequently re-infused into the patient’s body. This method uses chimeric antigen receptors and T cell receptors [27]. Currently, the National Cancer Institute and collaborators are conducting a phase I/II trial in cervical cancer patients; with the conclusion of enrollment in December 2018, we are awaiting the results from this study (NCT01583686).

### 5.2. Oncolytic Virotherapy

Oncolytic virotherapy, which uses the virus’ natural cytotoxic effects and immunostimulatory potential, has been attracting attention. This is a general term for treatment methods that use a virus that is cytotoxic against tumor cells and that is equipped with the capability to specifically infect cancer cells and use their machinery for proliferation. The virus itself generally exhibits cytotoxicity by infecting and proliferating in specific cells; oncolytic virotherapy uses viruses that possess the ability to more efficiently act on tumor cells or artificial viruses equipped with such characteristics through genetic modification techniques [28]. Up to the present time, researchers have conducted clinical trials using various viruses such as the herpes virus, adenovirus, and measles virus. Viral therapy not only directly exhibits anti-tumor effects, but also stimulates the immune system of the cancer patient to trigger an anti-tumor effect and is, therefore, considered to be one type of immunotherapy [29]. In patients with advanced and recurrent malignant melanoma, IMLYGIC^TM^ (Amgen Inc., Thousand Oaks, CA, USA), a genetically modified herpes virus, led to favorable outcomes in a phase III trial, resulting in FDA approval [30]. In cervical cancer, the development of viral therapies using the herpes virus is currently underway [31,32]. Although the clinical effects are yet unknown, its application in cervical cancer treatment is expected in the future.

**Table 1 healthcare-07-00108-t001:** Clinical research outcomes of immunotherapy in cervical cancer.

Study	Authors	Study Population	n	Phase	Treatment	Response	Toxicity
REGN2810	Papadopoulos et al., 2016 [33]	Advanced solid tumors	58	I	Cemiplimab monotherapy or combination with hfRT	DCR 62.8%	No dose-limiting toxicities
Keynote 028	Frenel et al., 2017 [22]	Recurrent cervical cancer with PD-L1 positive tumors	24	IB	Pembrolizumab	ORR 17% Median duration of response: 19.3 weeks 6-month PFS: 21%, OS: 66.7%	Grade 3 AE Rash and proteinuria
Keynote 158	Schellens et al., 2017 [23]	Recurrent cervical cancer	46	II	Pembrolizumab	ORR 12.2% (87% PD-L1+) >27 weeks follow-up: ORR: 27%	Grade 3 AE AST/ALT elevation and pyrexia
Checkmate 358	Hollebcque et al., 2017 [24]	Recurrent or metastatic HPV-related cancers	19	I/II	Nivolumab	ORR: 26.3% DCR: 70.8% Median PFS: 5.5 mo, OS: not reached	Grade 3–4 AE hyponatremia, syncope and diarrhea
	Lheureux et al., 2015 [34]	Recurrent or metastatic disease	42	I/II	Ipilimumab	Median PFS: 2.5 mo	Grade 3 AE Colitis and diarrhea
GOG9929	Mayadev et al., 2017 [35]	FIGO IB2/IIA or IIB/IIIB/IVA, positive nodes	34	I	CCRT with Ipilimumab	1 year DFS: 74%	Grade 1–2 AE Rash, Gastrointestinal toxicity

DCR: disease control rate; ORR: objective response rate; PFS: progression-free survival; OS: overall survival rate; AE: adverse event; and DFS: disease-free survival.

**Table 2 healthcare-07-00108-t002:** Ongoing clinical trials of immunotherapy for cervical cancer (monotherapy).

Clinical Trial Code	Title	Study Population	n	Phase	Treatment	Primary Outcomes	Secondary Outcomes
NCT02257528	Nivolumab in Treating Patients with Persistent, Recurrent, or Metastatic Cervical Cancer (NRG-GYO-02) [36]	Recurrent or metastatic cervical cancer	25	II	Nivolumab	ORR	PFS, OS
NCT03257267	Study of REGN2810 in Adults with Cervical Cancer (GOG 3016/ENGOT-cx9) (EMPOWER-Cervical) [37]	Recurrent or metastatic platinum-refractory cervical cancer	436	III	Cemiplimab	OS	PFS, ORR, DOR, QOL
NCT03972722	Study of GLS-010 Injection in Patients with Recurrent or Metastatic Cervical Cancer [38]	Recurrent or metastatic cervical cancer	89	II	GLS-010 (anti-PD1 antibody)	ORR	PFS, DCR, DOR, OS
NCT03104699	Phase 1/2 Study of AGEN2034 in Advanced Tumors and Cervical Cancer [39]	Advanced cancer Cervical cancer	75	I/II	AGEN2034 (anti-PD1 antibody)	DLT, MTD	Cmax, AUC, PFS, OS, DOR
NCT03808857	A Study in Recurrent or Metastatic Cervical Cancer Patients With PD-L1 Positive Who Failed in Platinum-based Chemotherapy [40]	Recurrent or metastatic cervical cancer	80	II	GB226 (anti-PD1 antibody)	ORR	TTR, DCR, DOR, OS
NCT03676959	A Clinical Study of PD-L1 Antibody ZKAB001 (Drug Code) in Recurrent or Metastatic Cervical Cancer [41]	Recurrent or metastatic cervical cancer	15	I	ZKAB001	DLT	MTD, ORR, AUC
NCT01693783	Ipilimumab in Treating Patients with Metastatic or Recurrent Human Papilloma Virus-Related Cervical Cancer [42]	Recurrent or metastatic cervical cancer	44	II	Ipilimumab	AE, ORR	

ORR: objective response rate; PFS: progression-free survival; OS: overall survival rate; DOR: duration of response; QOL: quality of life; DCR: disease control rate; AE: adverse event; DLT: dose-limiting toxicity; MTD: maximum tolerated dose; Cmax: maximum plasma concentration; AUC: area under curve; and TTR: To time of response.

**Table 3 healthcare-07-00108-t003:** Ongoing clinical trials of immunotherapy for cervical cancer (combination with CCRT).

Clinical Trial Code	Title	Study Population	n	Phase	Treatment	Primary Outcomes	Secondary Outcomes
NCT03833479	TSR-042 as Maintenance Therapy for Patients With High-risk Locally Advanced Cervical Cancer After Chemo-radiation (ATOMICC) [43]	Stage IB/IIA/IIB/III/IVA cervical cancer with pelvic or PALN	132	II	CRT Maintenance TSR-042 (anti-PD-1 antibody)	PFS	AE, OS
NCT03144466	A Study of Pembrolizumab And Platinum With Radiotherapy in Cervix Cancer (PAPAYA) [44]	Recurrent or metastatic platinum-refractory cervical cancer	26	I	CRT with pembrolizumab	MTD, Efficacy	PFS, OS
NCT03298893	Nivolumab in Association With Radiotherapy and Cisplatin in Locally Advanced Cervical Cancers Followed by Adjuvant Nivolumab for up to 6 Months (NiCOL) [45]	Locally advanced cervical cancer	21	I	CRT with nivolumab	DLT	DFS, AE, ORR, PFS
NCT03738228	Atezolizumab Before and/or With Chemoradiotherapy in Immune System Activation in Patients With Node Positive Stage IB2, II, IIIB, or IVA Cervical Cancer [46]	Stage IB/IIA cervical cancer with PALN or IIB/III/IVA cervical cancer with pelvic or PALN	40	I	Atezolizumab with CRT Atezolizumab before CRT	Clonal expansion of T cell receptor beta	Correlation of PD-L1 expression, PFS, AE, DLT
NCT02635360	Pembrolizumab and Chemoradiation Treatment for Advanced Cervical Cancer [47]	Locally advanced cervical cancer	88	II	Pembrolizumab with CRT	Change in immunologic markers following combination of study drug with chemoradiation, DLT	Metabolic Response Rate on PET/CT imaging, incidence of distant metastases, PFS, OS
NCT03612791	Trial Assessing the Inhibitor of Programmed Cell Death Ligand 1 (PD-L1) Immune Checkpoint Atezolizumab (ATEZOLACC) [48]	Locally advanced cervical cancer	190	II	Atezolizumab with CRT and adjuvant atezolizumab	PFS	
NCT01711515	Chemoradiation Therapy and Ipilimumab in Treating Patients With Stages IB2-IIB or IIIB-IVA Cervical Cancer [49]	Locally advanced cervical cancer	34	I	Ipilimumab with CRT	MTD, DLT	PFS

CRT: chemoradiation; PFS: progression-free survival; AE: adverse event; OS: overall survival rate; MTD: maximum tolerated dose; DLT: dose-limiting toxicity; DFS: disease-free survival; and ORR: objective response rate.

**Table 4 healthcare-07-00108-t004:** Ongoing clinical trials of immunotherapy for cervical cancer (combination with chemotherapy).

Clinical Trial Code	Title	Study Population	n	Phase	Treatment	Primary Outcomes	Secondary Outcomes
NCT03912415	Efficacy and Safety of BCD-100 (Anti-PD-1) in Combination With Platinum-Based Chemotherapy With and Without Bevacizumab as First-Line Treatment of Subjects With Advanced Cervical Cancer (FERMATA) [50]	Advanced cervical cancer	316	III	Paclitaxel + cisplatin (or carboplatin) Bevacizumab BCD-100 (anti-PD-1)	OS	PFS, ORR, DOR
NCT03912402	Efficacy and Safety of BCD-100 (Anti-PD-1) in Combination With Platinum-Based Chemotherapy and Bevacizumab in Patients With Recurrent, Persistent or Metastatic Cervical Cancer (CAESURA) [51]	Recurrent or metastatic cervical cancer	49	II	Paclitaxel + Carboplatin Bevacizumab BCD-100 (anti-PD-1)	ORR	PFS, OS
NCT03635567	Efficacy and Safety Study of First-line Treatment With Pembrolizumab (MK-3475) Plus Chemotherapy Versus Placebo Plus Chemotherapy in Women With Persistent, Recurrent, or Metastatic Cervical Cancer (MK-3475-826/KEYNOTE-826) [52]	Recurrent or metastatic cervical cancer	600	III	Paclitaxel + cisplatin (or carboplatin) Bevacizumab Pembrolizumab	PFS, OS	ORR, DCR, DOR
NCT03228667	QUILT-3.055: A Study of ALT-803 in Combination With PD-1/PD-L1 Checkpoint Inhibitor in Patients With Advanced Cancer [53]	Advanced cancer	611	II	ALT-803 (IL-15 superagonist) Pembrolizumab Nivolumab Atezolizumab Avelumab	ORR	PFS, OS, QOL
NCT03556839	Platinum Chemotherapy Plus Paclitaxel With Bevacizumab and Atezolizumab in Metastatic Carcinoma of the Cervix [54]	Recurrent or metastatic cervical cancer	404	III	Paclitaxel + cisplatin Bevacizumab Atezolizumab	OS	PFS, ORR, DOR, AE

OS: overall survival rate; PFS: progression-free survival; ORR: objective response rate; DOR: duration of response; DCR: disease control rate; QOL: quality of life; and AE: adverse event.

**Table 5 healthcare-07-00108-t005:** Ongoing clinical trials of immunotherapy for cervical cancer (combinations with checkpoint inhibitors).

Clinical Trial Code	Title	Study Population	n	Phase	Treatment	Primary Outcomes	Secondary Outcomes
NCT03894215	Phase 2 Study of Anti-PD-1 Independently or in Combination With Anti-CTLA-4 in Second-Line Cervical Cancer [55]	Recurrent cervical cancer	200	II	AGEN1884 (anti-PD-1) AGEN2034 (anti-CTLA4)	ORR	AE, DOR
NCT03495882	Subjects With Metastatic or Locally Advanced Solid Tumors, and Expansion Into Select Solid Tumors (Cervical) [56]	Cervical cancer	60	I/II	AGEN1884 (anti-PD-1) AGEN2034 (anti-CTLA4)	AE, DLT	ORR, DOR
NCT03972722	Study of GLS-010 Injection in Patients With Recurrent or Metastatic Cervical Cancer [38]	Recurrent or metastatic cervical cancer	89	II	GLS-010 (anti-PD1 antibody)	ORR	PFS, DCR, DOR, OS

ORR: objective response rate; AE: adverse event; DOR: duration of response; DLT: dose-limiting toxicity; PFS: progression-free survival; DCR: disease control rate; and OS: overall survival rate.

**Table 6 healthcare-07-00108-t006:** Ongoing clinical trials of immunotherapy for cervical cancer (combinations with molecularly-targeted therapy).

Clinical Trial Code	Title	Study Population	n	Phase	Treatment	Primary Outcomes	Secondary Outcomes
NCT03816553	SHR-1210 in Combination With Apatinib in Patients With Metastatic, Persistent, or Recurrent Cervical Cancer [57]	Recurrent or metastatic cervical cancer	49	II	SHR-1210 (anti-PD1 antibody) Apatinib (TK inhibitor)	ORR	PFS, DCR, DOR, OS
NCT02921269	Atezolizumab and Bevacizumab in Treating Patients With Recurrent, Persistent, or Metastatic Cervical Cancer [58]	Recurrent or metastatic cervical cancer	22	II	Atezolizumab Bevacizumab	Anti-tumor activity	OS, PFS, AE

ORR: objective response rate; PFS: progression-free survival; DCR: disease control rate; DOR: duration of response; OS: overall survival rate; and AE: adverse event.

## 6. Conclusions

Cancer immunotherapy is finally playing its role as a new therapeutic option for multiple types of cancer, giving new hope to patients with recurrent cancer. Nonetheless, the evidence is still insufficient in cervical cancer. While waiting for the results of currently ongoing trials, it is also necessary to narrow down the patients who could be expected to have satisfactory effects using good biomarkers and to consider combination therapy with existing treatments. To understand and develop such therapy, further studies of the tumor immune microenvironment and immune-related genes through polymorphism analysis are needed.

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
