# Peer review of "Immunotherapy for Uterine Cervical Cancer"

_healthcare, 2019, doi:10.3390/healthcare7030108_

Round 1
Reviewer 1 Report
I read with great interest the Manuscript titled “Immunotherapy for uterine cervical cancer” (healthcare-571474).
The topic of this manuscript falls within the scope of Healthcare.
I was particularly pleased to review this paper. In my honest opinion, the topic is interesting enough to attract the readers’ attention. The manuscript is a well written narrative review, and conclusions are supported by the reported evidence. Nevertheless, authors should clarify some point and improve the discussion.
In general, the Manuscript may benefit from several minor revisions, as suggested below:
All the text needs a minor language revision in order to improve some typos. In section 4.1 references are missed. In the first part of section 4.2 the references are missed. To date, several lines of evidence support the possibility to use specific biomarkers to identify early stage cervical cancer and, in this way, offer a better prognosis to the patients. I would suggest improving this manuscript at least briefly discussing this point (PMID: 28918603; PMID: 30579259)Author Response
Reviewer 1
In section 4.1 references are missed.
→Thank you for your comments. I added references.
In the first part of section 4.2 the references are missed.
→Thank you for your comments. I added references.
To date, several lines of evidence support the possibility to use specific biomarkers to identify early stage cervical cancer and, in this way, offer a better prognosis to the patients.
I would suggest improving this manuscript at least briefly discussing this point (PMID: 28918603; PMID: 30579259)
→Thank you for your comments. I added sentence of this point (Line 104 ~ 112).

Reviewer 2 Report
This review manuscript entitled ‘Immunotherapy for uterine cervical cancer’ is mainly described about clinical trial of immune checkpoint inhibitors and other types of immunotherapy. However, some incorrect data in clinical trials was found and the type of immunotherapy described in this manuscript is limited.
Six-month PFS in KEYNOTE-028 is not 13% but 21%. ORR in KEYNOTE-158 was not 17% but 12.2% in total population. Clinical response was observed only in PD-L1 positive cases. Therefore, drug effects cannot be independent of PD-L1 expression. It should be described that FDA concurrently approved PD-L1 IHC 22C3 PharmDx as a companion diagnostic. Since many clinical trials of immunotherapy targeting HPV related gene in cervical cancer, these are indispensable.
Author Response
Six-month PFS in KEYNOTE-028 is not 13% but 21%. ORR in KEYNOTE-158 was not 17% but 12.2% in total population. Clinical response was observed only in PD-L1 positive cases.
→Thank you for your comments. I corrected the description.
Therefore, drug effects cannot be independent of PD-L1 expression. It should be described that FDA concurrently approved PD-L1 IHC 22C3 PharmDx as a companion diagnostic.
→Thank you for your comments. I added sentence of this point
Since many clinical trials of immunotherapy targeting HPV related gene in cervical cancer, these are indispensable.
→Thank you for your comments. I added sentence of this point (Line 42 ~ 69).

Round 2
Reviewer 2 Report
The authors revised their manuscripts according to the comments of the reviewer. It is considered to be acceptable for the publication.